# Dysfunction of an Anaphase-Promoting Complex Subunit 8 Homolog Leads to Super-Short Petioles and Enlarged Petiole Angles in Soybean

**DOI:** 10.3390/ijms241311024

**Published:** 2023-07-03

**Authors:** Yi Gao, Jinlong Zhu, Hong Zhai, Kun Xu, Xiaobin Zhu, Hongyan Wu, Wenjing Zhang, Shihao Wu, Xin Chen, Zhengjun Xia

**Affiliations:** 1Key Laboratory of Soybean Molecular Design Breeding, Northeast Institute of Geography and Agroecology, Innovative Academy of Seed Design, Chinese Academy of Sciences, Harbin 150081, China; 2University of Chinese Academy of Sciences, Beijing 100049, China; 3Institute of Industrial Crops, Jiangsu Academy of Agricultural Sciences, Nanjing 210014, China

**Keywords:** short petiole, petiole angle, pulvinus, soybean, gibberellin, BVF-IGV

## Abstract

Plant height, petiole length, and the angle of the leaf petiole and branch angles are crucial traits determining plant architecture and yield in soybean (*Glycine max* L.). Here, we characterized a soybean mutant with super-short petioles (SSP) and enlarged petiole angles (named *Gmssp*) through phenotypic observation, anatomical structure analysis, and bulk sequencing analysis. To identify the gene responsible for the *Gmssp* mutant phenotype, we established a pipeline involving bulk sequencing, variant calling, functional annotation by SnpEFF (v4.0e) software, and Integrative Genomics Viewer analysis, and we initially identified *Glyma.11G026400*, encoding a homolog of Anaphase-promoting complex subunit 8 (APC8). Another mutant, *t7*, with a large deletion of many genes including *Glyma.11G026400*, has super-short petioles and an enlarged petiole angle, similar to the *Gmssp* phenotype. Characterization of the *t7* mutant together with quantitative trait locus mapping and allelic variation analysis confirmed *Glyma.11G026400* as the gene involved in the *Gmssp* phenotype. In *Gmssp*, a 4 bp deletion in *Glyma.11G026400* leads to a 380 aa truncated protein due to a premature stop codon. The dysfunction or absence of *Glyma.11G026400* caused severe defects in morphology, anatomical structure, and physiological traits. Transcriptome analysis and weighted gene co-expression network analysis revealed multiple pathways likely involved in these phenotypes, including ubiquitin-mediated proteolysis and gibberellin-mediated pathways. Our results demonstrate that dysfunction of *Glyma.11G026400* leads to diverse functional consequences in different tissues, indicating that this APC8 homolog plays key roles in cell differentiation and elongation in a tissue-specific manner. Deciphering the molecular control of petiole length and angle enriches our knowledge of the molecular network regulating plant architecture in soybean and should facilitate the breeding of high-yielding soybean cultivars with compact plant architecture.

## 1. Introduction

In soybean (*Glycine max*), the stem growth habit, i.e., indeterminate, determinate, or semi-determinate, greatly influences plant architecture. These growth habit types are largely controlled by the *Dt1* and *Dt2* genes [1], as well as maturity genes, e.g., *E1* and *Qne1* (*QTL* near *E1*) [2,3]. Plant canopy architecture is also influenced by leaf size, petiole length, and leaf petiole angle. Both light interception in the canopy and photosynthetic efficiency (and, thus, yield) can be enhanced by breeding for ideal canopy architecture [4,5]. In particular, short petiole length is desirable to soybean breeders since it allows for denser planting, thus increasing the number of plants per acre [6,7,8]. Several genes and loci controlling petiole length in soybean have been reported. Gao et al. (2022) characterized a mutant with short petioles and short plant height but an increased number of effective branches and a growth period prolonged by 2–7 days [9]. Kilen (1983) reported that the short vs. long petiole trait is controlled by the recessive gene *lps* [10]. You et al. (1998) reported that short petioles and an abnormal pulvinus are controlled by two recessive loci, *lps1* and *lps2*, respectively, by analyzing two mutant lines, NJ90L-1SP and D76-1609 [11]. Another study using soybean breeding line SS98206SP determined that the short petiole trait is controlled by *lps3* located in a 12 cM region between markers Sat_234 and Sct_033 in linkage group F (chromosome 13) [7]. Moreover, Liu et al. (2019) mapped two recessive loci, *dsp1* and *dsp2*, underlying the short-petiole phenotype to two nonhomologous regions of chromosome 7 and chromosome 11, respectively; 36 and 33 genes were predicted within the physical genomic interval of *dsp1* and *dsp2*, respectively [12].

Genes and loci controlling branch and petiole angle in soybean have also been identified. Clark et al. (2022) mapped a major quantitative trait locus (QTL) influencing branch angle, designated *qGmBa1*, to chromosome 19 using several types of mapping populations [13]. Additionally, Zhang et al. (2022) demonstrated that *PINFORMED1* (*GmPIN1*), encoding an auxin efflux transporter, determines polar auxin transport and regulates leaf petiole angle in an asymmetrical fashion in soybean [14].

More broadly, plant architecture is influenced by basic aspects of the eukaryotic cell cycle. The anaphase-promoting complex/cyclosome (APC/C), a multi-subunit E3 ubiquitin ligase, plays an important role in eukaryotic cell-cycle progression. Regulation of the APC/C during meiosis utilizes both mitotic APC/C regulators and meiosis-specific regulators [15]. The availability of APC/C subunits in specific plant tissues and/or cellular compartments might play an important role in regulating the APC/C. In rice (*Oryza sativa*), the *MONOCULM 1* (*MOC1*) gene is the key regulator controlling tiller number. *Tillering* and *Dwarf 1* (*OsTAD1*) encodes a coactivator of the APC/C [16]. TAD1 and OsAPC10 form a complex that functions as a coactivator of APC/C to target MOC1 for degradation in a cell-cycle-dependent manner [16].

*Increased Leaf Petiole Angle 1* (*GmILPA1*), encoding an anaphase-promoting complex 8 (APC8)-like protein, controls leaf petiole angle in soybean [17]. In cotton (*Gossypium hirsutum*), petiole angle is conferred by the uneven growth of cortex parenchyma cells on the adaxial and abaxial sides of the junction between the leaf blade and leaf petiole [18]. *GhAPC8*-silenced plants exhibited reduced plant height and leaf blade angle and contained higher levels of brassinosteroid (BR) and lower levels of auxin and gibberellin (GA) at this junction compared to the wildtype [18]. Comparative transcriptome analysis revealed that silencing *GhAPC8* activated BR biosynthesis and signaling pathway genes, as well as genes related to ubiquitin-mediated proteolysis [18].

In the current study, to explore the molecular basis of the control of leaf petiole length and leaf petiole angle in soybean, we characterized a soybean mutant with super-short petioles and enlarged petiole angles, which we named *Gmssp*. We identified *Glyma.11G026400* as the candidate gene responsible for the *Gmssp* phenotype using a pipeline involving manual bulk sequencing, variant calling, functional annotation by SnpEFF software, and Integrative Genomics Viewer analysis (BVF-IGV). We validated our findings by QTL mapping and allelic variation analysis and by characterizing another *Glyma.11G026400* mutant with a similar phenotype to *Gmssp*. Furthermore, anatomical structure analysis, transcriptome analysis, and weighted gene co-expression network analysis (WGCNA) revealed that several molecular pathways are involved in the *Gmssp* phenotype, including APC-mediated proteolysis and the GA and BR pathways. Our findings should help breeders create new, high-performing soybean cultivars with improved plant architecture and higher yield potential.

## 2. Results

### 2.1. The Gmssp Mutant Has a Dwarf Phenotype with Super-Short Petioles and Enlarged Petiole Angles

We looked for the *Gmssp* phenotype in the M_3_ (mutant population generation 3) generation of a mutant library of soybean cultivar Heihe 43 obtained by cobalt (^60^Co) gamma ray irradiation. Of the 10 plants of line 399, four plants displayed the *Gmssp* phenotype, with super-short petioles and enlarged petiole angles, whereas the other six plants exhibited a wildtype (WT) phenotype (Figure 1A–F). Line 399 was derived from a single M_2_ plant, indicating that the *Gmssp* locus was heterozygous in the M_2_ generation.

In the M_4_ generation, all *Gmssp* mutant progeny displayed *Gmssp* phenotypes. Of the six WT-derived lines examined, three displayed the WT phenotype and three showed the *Gmssp* phenotype. In the progeny of the three WT lines, 39 out of 227 plants displayed the *Gmssp* phenotype. A chi-square test revealed that all three lines fit the 3:1 Mendelian segregation ratio for a single recessive gene at *p* = 0.1100 to 0.4781.

The average petiole length in the M_4_ population was 2.68 ± 0.52 cm in the *Gmssp* mutants and 19.51 ± 5.20 cm in the WT; the average petiole length of *Gmssp* was 13.74% that of the WT (*p* < 0.001; Figure 1G). *Gmssp* also had significantly larger angles between the main stem and petiole (106.51 ± 19.78°) than the WT (34.86 ± 3.05°) (*p* < 0.001; Figure 1H). Furthermore, *Gmssp* plants were significantly shorter (68.02 ± 2.45 cm) than WT plants (108.92 ± 2.96 cm) (*p* < 0.001; Figure 1I) and had significantly smaller leaves (19.11 ± 4.0 cm^2^) than WT plants (33.02 ± 3.05 cm^2^) (*p* < 0.001; Figure 1D,E). The 100-seed weight of *Gmssp* plants (19.70 ± 0.81 g) was lower than that of WT plants (20.83 ± 1.07 g) (*p* = 0.015; Appendix A). Furthermore, *Gmssp* had a lower oil content (18.10 ± 0.72%) and higher protein content (42.63 ± 0.15%) than the WT (20.50 ± 0.44% for oil content and 41.40 ± 0.40% for protein content) (*p* < 0.01; Appendix A). Furthermore, *Gmssp* had a lower lignin content (6.78 ± 0.09%) than the WT (8.85 ± 0.11%) at *p* < 0.01 (Figure 1K). In addition, *Gmssp* had lower neutral detergent fiber (NDF), acid detergent fiber (ADF), and crude fiber (CF) contents than the WT (Appendix A).

We investigated the anatomical differences between *Gmssp* and WT tissues. In transverse sections, *Gmssp* petioles were smaller in diameter than WT petioles. The vascular bundles were well organized in the WT, but not in *Gmssp*, in which they were replaced by undifferentiated cells (Figure 2A,B). In longitudinal sections of the petiole, we observed large, well-developed cells in the WT but cells that were small and compact, especially in length (Figure 2C,D), in *Gmssp*, which is in accordance with the super-short petioles of *Gmssp*. We did not observe many structural differences in cross-sections of leaves between the two genotypes (Figure 2I,J), and the *Gmssp* stem was similar to that of the WT except that some parenchymal cells were smaller in *Gmssp* (Figure 2E–H).

In the pulvinus, *Gmssp* displayed a large area of undifferentiated cells, leading to an underdeveloped pulvinus compared to the WT (Figure 2K,L). This structural difference between *Gmssp* and the WT is similar to that previously described for a mutant of *GmILPA1* [17]. These anatomical defects in the *Gmssp* pulvinus are expected to alter its ability to bear the weight of the leaf and petiole, leading to an enlarged petiole angle.

### 2.2. The BVF-IGV Pipeline Identified Glyma.11G026400 as the Candidate Gene for the Gmssp Phenotype

We extracted two pools of genomic DNA from leaf tissues of four *Gmssp* plants and six WT plants of line 399. The samples were subjected to 150 bp paired-end resequencing on an Illumina platform at Annoroad Gene Technology (Beijing, China). For the *Gmssp* mutant pool, the total number of reads was 239,532,532, the number of high-quality (HQ) reads was 233,689,676, and the total number of bases was 35,929,879,800. For the WT pool, the total number of reads was 205,505,624, the number of HQ reads was 200,020,114, and the total number of bases was 30,825,843,600.

Initially, we processed the resequencing data using MutMap (v1.4.4) software [19]; the results are shown in Appendix A. However, there were many peaks spanning relatively large genomic regions of several chromosomes, e.g., chromosomes 5, 6, and 11. Although large genomic regions could be excluded, the large number of peaks or regions made it difficult to rule them out one by one. The heterozygosity of the soybean genome may contribute to the presence of multiple peaks or regions when using the default settings in MutMap. Hence, we developed the BVF-IGV pipeline as a means to identify the causal mutation underlying the *Gmssp* phenotypes (Figure 3A).

After the reads were quality-trimmed using the NGS QC Toolkit with default parameters, we aligned the bulk sequences of four *Gmssp* mutants and six WT plants to the reference genome Gmax_275_Wm82.a2.v1 (V275) using BWA (v0.7.13) software [20]. Alternatively, we analyzed clean reads using SpeedSeq (v0.1.2) software [21,22]. We used GATK v.2.3-3 software to call variants from BAM files using the parameters -stand_emit_conf 10 and -stand_call_conf 30. We filtered the VCF file using the FilterVcf function in Picard (v2.1.1) software with the parameters MIN_AB = 0.8, MIN_DP = 6, MIN_GQ = 0, and MIN_QD = 2. The filtered VCF file was subjected to functional annotation via SnpEFF software [23] with default parameters (Figure 3A).

We included only allelic variations annotated as a “missense variation” or “frameshift variation” in the first round of analysis. This identified 121,763 allelic variations for the *Gmssp* bulk and 122,991 for the WT bulk compared with the V275 reference genome. By manipulating the VCF data in Excel, we eliminated the common allelic variations between the *Gmssp* and WT bulks, leaving 3964 loci (1760 genes) specific to *Gmssp* and 4922 loci (2030 genes) specific to the WT. We manually checked these genes individually or in batches by taking snapshots of each gene into a fold using the “run batch script” function in IGV. The numbers of genes on each chromosome are listed in Appendix A. The numbers of *Gmssp-* and WT-specific genes on each chromosome were highly correlated, but they were not correlated to the lengths of the chromosomes. After an initial check, we identified a 4 bp deletion in *Glyma.11G026400*, marking this gene as a strong candidate for *Gmssp*. Notably, the *Gmssp* bulk showed homogenous deletions for all reads, whereas approximately one in six reads of the WT bulk contained the 4 bp deletion. This observation is in good accordance with the finding that half of WT plants were heterozygous at *Gmssp* (Figure 3B).

We ruled out all other genes because the heterozygous ratios at each allelic variant between *Gmssp* and the WT did not fit the prediction. We also investigated the 2030 WT-specific genes, since large deletions in the *Gmssp* genome might be captured by observing WT-specific genes. Taken together, a 4 bp deletion of *Glyma.11G026400* was identified as the functional mutation for *Gmssp* using the BVF-IGV pipeline.

### 2.3. QTL Mapping Validates Glyma.11G026400 as the Candidate Gene for Gmssp

To further test whether *Glyma.11G026400* was the gene that harbored the *Gmssp* mutation, we crossed the *Gmssp* mutant with Hefeng 55 and selfed the F_1_ plants to generate F_2_ populations for QTL mapping. We generated three F_2_ subpopulations from individual F_1_ plants. In these three subpopulations, 53 out of 144, 36 out of 177, and 38 out of 183 individuals displayed the *Gmssp* phenotype. According to a chi-square test, the segregation patterns for all three subsets fit the expected 3:1 Mendelian segregation ratio for a single recessive gene at *p* = 0.661.

We identified polymorphisms (Indels and SNPs) on the basis of the resequencing data of the parents, *Gmssp* (in the Heihe 43 background) and Hefeng 55, using newly developed Indels and SNPs markers in the region where the candidate gene is anchored. We then performed QTL analysis for *Gmssp* (super-short petioles and enlarged petiole angle) (Appendix A), which showed that the QTL peak overlapped with the candidate gene-anchored region (Figure 3C). We analyzed the allelic variations among *Gmssp*, WT (Heihe 43 background), and Hefeng 55 within the QTL peak region, finding that a 4 bp deletion leading to a premature stop codon in *Glyma.11G026400* in *Gmssp* was the only mutation in this region (Appendix A). These QTL mapping results strongly support the results using our BVF-IGV pipeline (Appendix A).

### 2.4. Characterization of the t7 Mutant Supports Glyma.11G026400 as the Causal Gene for the Gmssp Phenotype

In the M_3_ generation of a Dongsheng 7 mutant library, one mutant, *t7*, displayed super-short petioles and an enlarged petiole angle, similar to the *Gmssp* phenotype (Figure 4A–G). Furthermore, F_2_ plants from a cross of *t7* and WT plants from the same background (Dongsheng 7) fit a 3:1 Mendelian segregation ratio for a single recessive gene. The *t7* mutant was much shorter (33.43 ± 5.42 cm) than the WT (86.50 ± 2.07 cm) and contained fewer nodes (16.87 ± 2.28) than the WT (19.60 ± 0.52) (*p* < 0.001 for both) (Figure 4D,E). The average petiole length of *t7* was 1.75 ± 0.26 cm, only ~13% of the WT petiole length (16.31 ± 1.63 cm) (*p* < 0.001; Figure 4F), and *t7* had a significantly larger petiole angle (100.35 ± 12.75°) than the WT (33.71 ± 12.75°) (*p* < 0.001; Figure 4G).

We performed RNA sequencing (RNA-seq) and 150 bp paired-end resequencing of genomic DNA from a mutant bulk of 20 *t7* plants and a WT bulk of 20 plants (Appendix A). For RNA-seq, we detected no expression for 41 genes within the *Glyma.11G026400*-anchored region in the *t7* bulk, suggesting that a large deletion occurred in this region (Appendix A). We employed the primers F85, F399, and F20 used for QTL mapping and newly designed primers (T7-1 and T7-5) using Primer3 (v2.5.0) software to validate the existence of this deletion (Chr11:1695723 to Chr11:1995601) (Appendix A). The PCR results confirmed the lack of gene expression in this region in *t7* (Figure 4H).

We analyzed the resequencing data of the *t7* and WT bulks, which further confirmed the deletion (Figure 4F). Using IGV, we clearly observed the deletion running from Chr11:1695723 to Chr11:1995601, within which 41 genes (*Glyma.11G023700* to *Glyma.11G027700*; splicing variants were not included) were annotated. The candidate gene *Glyma.11G026400* was located within the deleted region (Figure 4I–K). The similar phenotypes of *t7* and *Gmssp* support the notion that *Glyma.11G026400* is the causal gene for the *Gmssp* phenotype and that the protein encoded by *Glyma.11G026400* in *Gmssp* might be dysfunctional.

### 2.5. Glyma.11G026400 Encodes a Truncated Protein in Gmssp

*Glyma.11G026400* was functionally annotated to be *ANAPHASE-PROMOTING COMPLEX SUBUNIT 8* (APC8/CDC23), which produces a 2087 bp transcript and contains a 1734 bp coding sequence encoding a 577 amino-acid (aa) protein (Gmax_275_Wm82.a2.v1 (V275)). This gene is annotated in the Gene Ontology (GO) database with the terms GO:0005515 (protein binding), GO:0030071 (regulation of mitotic metaphase/anaphase transition), and GO:0005680 (anaphase-promoting complex). Here, we identified a 4 bp deletion in the third exon of *Glyma.11G026400* in the *Gmssp* mutant, leading to a premature stop codon and encoding a truncated protein of 380 aa, i.e., 197 aa shorter than the WT protein (Figure 5A–C).

To obtain functional clues on the basis of the conservation and diversification of homogenous sequences, we generated a phylogenetic tree using APC8 homologs from soybean and closely related legume species, as well as the model crops rice and *Arabidopsis* (*Arabidopsis thaliana*) (Figure 5D). The APC8 protein sequences most closely related to soybean were those of other leguminous species. In *Arabidopsis* (TAIR10 assembly), AtAPC8 (*AT3G48150*) shares 89.6% sequence similarity with GmAPC8 encoded by *Glyma.11G026400* (Figure 5D). The predicted 3D protein structures of GmAPC8 and Gmapc8 from the WT (Figure 5E) and the *Gmssp* mutant (Figure 5F), respectively, differed markedly due to the truncation of 197 aa in Gmapc8.

### 2.6. The Mutation in Gmapc8 in the Gmssp Mutant Does Not Alter Its Subcellular Localization

We conducted subcellular localization analysis of GmAPC8 and Gmapc8 proteins from the WT and *Gmssp*, respectively, in lower epidermal cells of *Nicotiana benthamiana* leaves. Both of these proteins localized to the nucleus and chloroplasts (Appendix A). Thus, the mutation in Gmapc8 did not alter their subcellular localization.

### 2.7. DEGs Are Enriched in Ubiquitin-Mediated Proteolysis and Hormone Signal Transduction

To explore the molecular mechanism of *Gmssp*, we performed RNA-seq of petiole, pulvinus, leaf, stem, and apical meristem tissues and analyzed the data using the RNA-seq workflow pipeline (Appendix A). We manually checked the authenticity of the genotype at the *GmAPC8* locus for each sample using IGV.

The numbers of differentially expressed genes (DEGs) between the WT and *Gmssp* varied widely in different tissues (Figure 6A,B). In the petiole, 6606 genes were significantly differentially expressed. In Gene Ontology (GO) enrichment analysis of these petiole DEGs, 105 terms were enriched, with 84 KappaScore groups (Figure 6C). Significantly enriched pathways included taurine and hypotaurine metabolism (12 genes), MAPK signaling (59 genes), oxidative phosphorylation (51 genes), cysteine and methionine metabolism (43 genes), ubiquitin-mediated proteolysis (27 genes), and plant hormone signal transduction (90 genes).

We identified 391 significant DEGs in the pulvinus, which were significantly enriched in three pathways: phenylalanine metabolism, phenylpropanoid biosynthesis, and protein processing in the endoplasmic reticulum. In the stem, we detected 980 significantly DEGs and six enriched pathways, including phenylpropanoid biosynthesis, MAPK signaling, plant–pathogen interaction, and plant hormone signal transduction (14 genes). In the apical meristem, we identified 6054 DEGs and 86 enriched terms (specific for clusters) with 68 KappaScore groups (Appendix A). Five pathways were significantly enriched: ubiquitin-mediated proteolysis (50 genes), spliceosome (62 genes), ribosome (42 genes), RNA polymerase (25 genes), and plant hormone signal transduction (79 genes).

We identified 3562 and 11,160 DEGs between the WT and *Gmssp* in leaves at the V4 and V8 stages, respectively, whereas we detected 4645 DEGs in leaves at the V4 stage between *t7* and the WT (Figure 6). We identified 63–107 enriched GO terms (specific for clusters) in 50–85 KappaScore groups for these three leaf samples (Appendix A). Among the 63 enriched GO terms identified in leaf tissue at the V4 stage, photosynthesis-related pathways, i.e., photosynthesis (35 genes), photosynthesis_1 (14 genes), and porphyrin and chlorophyll metabolism (16 genes) were significantly enriched, as were nitrogen metabolism (13 genes), DNA replication (30 genes), ubiquitin-mediated proteolysis (10 genes), and plant hormone signal transduction (54 genes). Among the 63 GO terms identified in leaf tissue at the V8 stage, ribosome (299 genes), ubiquitin-mediated proteolysis (31 genes), and plant hormone signal transduction (152 genes) were enriched. Among the 79 GO terms identified in leaf tissue at the V4 stage between *t7* and the WT, proteasome (24 genes), nucleotide excision repair (26 genes), mismatch repair (27 genes), ubiquitin-mediated proteolysis (12 genes), and plant hormone signal transduction (66 genes) were significantly enriched.

Given the enrichment of the plant hormone signal transduction pathways in stem tissue and ubiquitin-mediated proteolysis in the other tissues (which is the function of APC8), we analyzed the DEGs involved in APC-related ubiquitin-mediated proteolysis as well as GA-, cytokinin (CK)-, auxin-, and BR-mediated signal transduction pathways. As shown in the heatmap (Figure 6D), six GA-related genes, an auxin-related gene (*Glyma.03G169600*), and 12 APC-related genes were significantly downregulated in the *Gmssp* mutant. Conversely, other APC-, GA-, CK-, and BR-related genes were upregulated in *Gmssp*.

As shown in the Venn diagrams in Appendix A, there were 56 petiole-specific APC-, GA-, CK-, auxin-, and BR-related DEGs between the WT and *Gmssp*, whereas there were only one and three DEGs specific for the pulvinus and stem, respectively. For leaf tissue, 20 common DEGs were shared between leaves at three growth stages and the apical meristem.

### 2.8. Gibberellin Treatment Partially Rescues the Phenotype of Gmssp

GAs are a class of tetracyclic diterpenoid phytohormones that regulate developmental processes such as leaf expansion, stem elongation, flower induction, and development to seed development. To investigate whether the phenotype of *Gmssp*, especially the short petioles, could be rescued by GA treatment, we treated plants with 0.1 μM GA_3_. GA treatment restored the plant height and petiole length, but not the petiole angle, of the *Gmssp* mutant (Appendix A). This phenomenon could be explained by the presence of GA pathway genes or a very low level of GA in the petiole. However, genes underlying the abnormality of the pulvinus might be independent of the GA pathway.

### 2.9. WGCNA Reveals Modules Associated with Traits Affecting Both Tissues and Growth Stages of Gmssp

We subjected all RNA-seq data and the corresponding phenotypic data to WGCNA (Appendix A). As observed in the module–trait relationships shown in Figure 7A, the MEsaddlebrown, MEred, MEbrown, and MEblack modules were significantly associated with the mutation at *Glyma.11G026400*. The MEsaddlebrown module was not related to tissues, with a correlation coefficient of –0.17 (*p* = 0.3), but this module appeared to be related to growth stage (correlation coefficient of 0.28), although this association was not statistically significant (*p* = 0.06) (Figure 7A).

When we analyzed the tissues and stages, we identified many modules strongly associated with both traits. Thirteen modules were significantly associated with the tissue trait at *p* < 0.01 and five modules were significantly associated at 0.05 > *p* > 0.01. MEgrey60 was significantly associated with the tissue trait, with a correlation coefficient of −0.6 at *p* = 1 × 10^−5^; MEmagenta was significantly associated with this trait, with a correlation coefficient of 0.57 at *p* = 4 × 10^−5^; MEyellow was significantly associated with this trait, with a correlation coefficient of 0.57 at *p* = 5 × 10^−5^ (Figure 7A). Only modules MEorange and MEdarkgrey were significantly associated with the stage trait, with correlation coefficients of 0.55 (*p* = 1 × 10^−4^) and 0.42 (*p* = 0.004), respectively (Figure 7A). Figure 7B shows the modules associated with mutations at *p* ≤ 0.1, as well as the number of genes involved in the APC, auxin, CK, BR, and GA pathways (Appendix A). Of all the listed pathways, the APC and GA pathways were the most highly enriched (Appendix A).

The genes of the MEsaddlebrown module are listed in Appendix A, and the hub gene network is presented in Appendix A. Since genes of MEsaddlebrown were not related to known pathways, genes or hub genes in the MEsaddlebrown module might underlie a different mechanism or function of APC8 (Appendix A).

## 3. Discussion

### 3.1. Short Petioles and Enlarged Leaf Petiole Angles Are Important Traits Determining Plant Architecture in Soybean

The relationship between plant architecture and yield in soybean is complex and dependent on the environment, cultivation method, and planting density. In general, short petioles and small petiole angles are desirable traits for ideal plant architecture to ensure a high yield potential. Researchers and breeders have reported QTLs for the short petiole trait, i.e., *lps1*, *lps2*, and *lps3* [7,10,11]; however, the detailed molecular mechanisms underlying this trait are still unclear. Gao et al. (2017) reported an APC8-like protein encoded by *GmILPA1* controlling petiole angle in soybean [17]. Here, we characterized a *Gmssp* mutant with super-short petioles and enlarged petiole angles. Notably, the gene identified in our study is the same one reported by Gao et al. (2017) [17], although the reported phenotypes differ. These differences may arise from a dose effect of *Glyma.11G026400* alleles [24], i.e., a weak allele (*Gmilpa1*) encoding a protein with partial function in the Gao et al. (2017) study versus the dysfunctional protein in the *Gmssp* mutant.

### 3.2. The BVF-IGV Pipeline Identifies the Causal Mutation for the Gmssp Phenotype

Bulked-segregant analysis (BSA) is widely used to identify the genetic variations associated with specific phenotypic traits [25]. Advances in sequencing technology, including next-generation sequencing and third-generation long-read sequencing of DNA and RNA bulks, provide sequence information quickly in a cost-effective manner. Furthermore, data processing software, e.g., MutMap [26], and improved pipelines [27,28], such as Mu-seq [29], BSAseq [30], and MMAPPR [31], can be used to identify the causal gene for a specific phenotype. In this study, we initially used MutMap to identify the candidate region or locus responsible for the *Gmssp* phenotype. However, we obtained too many peaks spanning a relatively large genomic region using default parameters in MutMap [19,26], and there were too many heterozygous loci that we could not easily eliminate. Hence, we developed the BVF-IGV pipeline. Segregation analysis enabled us to predict the ratio of the mutant allele in a WT bulk, which was crucial for accurately identifying the causal mutation. Typically, after filtering, a few hundred loci or genes associated with missense or frameshift mutations, according to annotation by SnpEFF [23], can be easily and thoroughly checked with the aid of IGV. In this study, we easily identified the candidate gene *Glyma.11G026400* out of 1760 *Gmssp*-specific genes since we could view every candidate gene on IGV directly and automatically or take snapshots for each gene before viewing by running a simple script. In general, we easily ruled out all genes except the true candidate gene based on their segregation ratios (Figure 3B). We validated our results by QTL mapping of the candidate region in the F_2_ population of a *Gmssp* × Hefeng 55 cross, followed by allelic variation analysis. Furthermore, analysis of the *t7* mutant—which contains a multi-gene deletion that includes the entire *Glyma.11G026400* sequence and has a similar phenotype to *Gmssp*, with super-short petioles and enlarged petiole angles—allowed us to further validate *Glyma.11G026400* as the causal gene for the *Gmssp* phenotype.

However, it might not be possible to easily identify causal mutations present in promoter regions or in regions with low read coverage from paired-end sequencing using the BVF-IGV pipeline. When “speedseq” is used for variant calling, structural variations, such as a large deletion or copy number variations, can be identified on IGV. Indeed, in a previous study, we identified over 30 causal genes for various mutations using the BVF-IGV pipeline in soybean and common bean (*Phaseolus vulgaris*) [22].

### 3.3. Glyma.11G026400 Affects Plant Architecture via Ubiquitin-Mediated Proteolysis (APC) and Plant Hormone Signal Transduction

The anaphase-promoting complex and cyclosome (APC/C) are components of a large multi-subunit E3-ubiquitin ligase complex that plays an essential role in proteasomal degradation during the cell cycle via ubiquitination of its target proteins [32]. The tetratricopeptide repeat (TPR) domain-containing protein APC8 and other APC family members, i.e., APC6 and APC7, form a structural arm within the APC/C complex and are essential for its ubiquitin ligase activity [33,34]. The regulation of the APC/C complex during meiosis is carried out by mitotic regulators of the APC/C, as well as meiosis-specific regulators [15]. In human colon epithelial cells, the expression of a truncation mutant of *APC8/CDC23* (*CDC23 ΔTPR*) led to abnormal levels of APC/C targets, such as cyclin B1, and disturbed cell-cycle progression via mitosis [35]. Temperature-sensitive mutations in subunits of the *C. elegans* APC arrest at the metaphase of meiosis I at the restrictive temperature. Embryos depleted of the APC coactivator FZY-1 by RNA interference also arrest at this stage. In *Arabidopsis AT3G48150* mutants, the loss of *APC8* or *CDC23* leads to defects in male gametogenesis [36,37], and a weak *APC8* allele identified in a mutant screen conferred similar abnormalities in inflorescence, leaf, and shoot meristem development [37]. These findings suggest that different forms of the APC may carry out different tasks during plant development due to divergent expression patterns among the genes encoding its subunits, suggesting that the roles of APC/C in plants may be influenced by subunit availability in specific tissues or cellular compartments [38].

In the current study, the dysfunction or deletion of *Glyma.11G026400* resulted in super short petioles and enlarged petiole angles, as well as other phenotypic and physiological changes, i.e., reduced plant height and lignin content, but did not affect reproductive development, e.g., flowering time or seed protein + oil content. Furthermore, the DEGs varied widely among different tissues, which might help explain why the APC8 homolog *Glyma.11G026400* plays key roles in cell differentiation and elongation in a tissue-specific manner.

Given that Gao et al. (2017) reported that the mutant protein in *Gmilpa1* (encoded by *Glyma.11G026400*) was partially functional [17], the Gmapc8 mutant protein in *Gmssp* (also encoded by *Glyma.11G026400*) lacking 197 aa might be nonfunctional, i.e., equivalent to the total deletion of this same gene in the *t7* mutant. Therefore, we demonstrated that the functions of protein variants encoded by different *Glyma.11G026400* alleles differ from the functions of proteins encoded by the *APC8* genes reported in *Arabidopsis* [36] and cotton [18], pointing to their functional divergence.

GmAPC8 directly interacts with GmAPC13a as part of the APC complex [16]. The functions of APC/C in plants may vary depending on the availability of different APC/C subunits in specific tissues and/or cellular compartments (Figure 8). The pulvinus had the fewest DEGs among the tissues examined, suggesting that the expression levels of most genes might be lower in the pulvinus than in other tissues. Therefore, some APC subunits might not be available in the *Gmssp* pulvinus. The dominant form of *Glyma.11G026400* might be needed for proper cell growth and division in the pulvinus. The current and previous studies showed that partially and fully dysfunctional GmAPC8 proteins result in enlarged petiole angles. The outer cells of the parenchyma, termed the “motor cells”, are responsible for nyctinastic and thigmonastic movements through water-driven changes in volume [39]. Reduced motor cell proliferation in the *Gmilpa1* and *Gmssp* mutants might be the physical mechanism underlying the enlarged petiole angles.

The MEsaddlebrown module, consisting of 50 genes, was highly associated with the mutation in *Glyma.11G026400* in the *Gmssp* mutant, but no known APC, GA, auxin, CK, or BR pathway genes were enriched in this module. These genes, especially the hub genes (Appendix A) identified by cytoHubba [40], might underlie a functional mechanism for GmAPC8, which merits further study.

## 4. Materials and Methods

### 4.1. Mutant Library Development, Phenotyping, and Analysis of Anatomical Structure

The soybean (*Glycine max*) cultivars Heihe 43, Dongsheng 7, and Hefeng 55 are cultivated in northern China. Mutant libraries were constructed by mutagenesis of the above cultivars using ^60^Co gamma rays [41]. In the M_3_ generation, plants derived from a single self-pollinated M_2_ plant were planted in the field for phenotypic observation. In line 399 (derived from Heihe 43), mutants with super-short petioles and enlarged petiole angles were referred to as *Gmssp*. Progeny from self-pollinated line 399 plants were harvested and advanced to the next generation to observe the segregation for the *Gmssp* phenotype, as well as other phenotypic traits. For phenotyping, the length of the leaf petiole in the main stem was measured with a ruler, whereas the angle between the main stem and petiole was measured with a protractor. To measure leaf area, the leaf was photographed along with horizontal and vertical rulers, and leaf area was measured using ImageJ (v1.53a) software. At the R7 stage, plant height was measured from the cotyledon node to the shoot tip of the main stem using a ruler. Seed protein and oil contents of each plant or genotype were measured using a MATRIX-I FTNIR spectrometer (Bruker, Billerica, MA, USA). Leaf, petiole, pulvinus, stem, and apical meristem tissues from *Gmssp* and WT plants were sampled for RNA-seq. Basic information on genotypes, tissues, and sampling times (at different growth stages) is provided in Appendix A.

To observe anatomical structures, plant samples were fixed in FAA fixative (5 mL of formaldehyde, 50 mL of ethanol, and 10 mL of glacial acetic acid, with a constant volume of 100 mL). Tissues from *Gmssp* and the WT were sampled and fixed in FAA fixative with four rounds of vacuum infiltration overnight at 4 °C. A series of treatments using ethanol, xylene, and Paraplast was performed as described previously [17]. The sections were cut on a RM2245 microtome (Leica, Wetzlar, Germany), stained with toluidine blue, and observed under a SZX6 microscope (Olympus, Tokyo, Japan). Lignin, neutral detergent fiber (NDF), acid detergent fiber (ADF), and crude fiber (CF) contents were determined using near-infrared spectroscopy as described [42]. The figures show a representative result. Data in all bar graphs represent the means ± standard deviation (SD). To compare phenotypic data between mutant and WT plants, Student’s *t*-test (*t* test), analysis of variance (ANOVA), and chi-square test were performed using GraphPad Prism version 5.00 for Windows, GraphPad Prism (v9.5.1) (San Diego, CA, USA).

### 4.2. Resequencing and Identification of the Causal Gene by MutMap and BVF-IGV

Genomic DNA was extracted from leaf tissues of four *Gmssp* plants and six WT plants of line 399 using leaves of the same size for each plant. Resequencing of genomic DNA was performed on an Illumina platform at Annoroad Gene Technology (Beijing, China). Reads were quality trimmed using the NGS QC Toolkit (v2.3.3) with default parameters [43]. The sequences of the two bulks were used to identify a causal region or peak using MutMap software with default parameters [19,26]. Due to the limitations of MutMap revealed in this study, a manual BVF-IGV pipeline was developed, consisting of bulk sequencing, variant calling, functional annotation by SnpEFF, and batch Integrative Genomics Viewer (IGV) observation to identify the candidate gene.

For each sample, only reads that passed the quality check as matching pairs were retained and aligned to the reference genome Gmax_275_Wm82.a2.v1 (V275) using the Burrows–Wheeler Alignment Tool (BWA) v.0.6.2 [20,44]. Alternatively, clean reads were analyzed using SpeedSeq software [21,22]. The resulting SAM files were converted to sorted BAM files compliant with the Genome Analysis Toolkit (GATK) format by Picard Tools v. 1.77 using the tools in the following order: CleanSam, SamFormatConverter, and AddOrReplaceReadGroups. GATK v.2.3-3 software was used to call variants from BAM files. The VCF files were subjected to filtering, functional annotation by SnpEFF, selection, and batch IGV observation [45]. Refer to Section 2 for details on the BVF-IGV pipeline.

### 4.3. Generation of the Mapping Population and Markers

To verify the BVF-IGV results, the homogenous *Gmssp* (male) in the M4 generation was crossed with Hefeng 55 (female) to generate a QTL mapping population. DNA was extracted from plants in the F2 population using the CTAB method [46]. Markers for QTL mapping in the putative candidate gene-anchored region were developed by targeting the polymorphisms between *Gmssp*/Heihe 43 and Hefeng 55 on the basis of the resequencing data of *Gmssp*, Heihe 43, and Hefeng 55 (Appendix A). In general, InDel variants were used to develop InDel markers, whereas SNP variants were used to develop CAPS/dCAPS markers. QTL IciMapping version 4.0 software [47] was used for QTL analysis.

### 4.4. RNA Sequencing and Transcriptome Analysis

The apical meristem, leaf, stem, petiole, and pulvinus tissue of *Gmssp* and the WT were sampled for RNA-seq (Appendix A). RNA library preparation (preparation of RNA samples, generation of sequencing libraries, and paired-end sequencing) was conducted as previously reported [2]. Sequencing was performed at Annoroad Gene Technology Corporation. The RNA sequencing data were analyzed through the RNAseq-workflow pipeline (https://github.com/twbattaglia/RNAseq-workflow, accessed on 10 November 2022) against the soybean reference genome (V275 of Wm82.a2.v1, phytozome-next.jgi.doe.gov, accessed on 1 March 2020).

Reverse-transcription quantitative PCR (RT-qPCR) was performed to validate the RNA-seq results. Total RNA was extracted from different tissues using an OminiPlant RNA Kit (DNase I) (CW25985, CEBIO). A 500 ng RNA sample was reverse-transcribed using TransScript One-Step gDNA Removal and cDNA Synthesis SuperMix (AT311-03, TransGen Biotech, Beijing, China). TransStart Top Green qPCR SuperMix (AQ131-04; TransGen Biotech, Beijing, China) was used for the qPCR assays. qPCR was conducted using a LightCycler 96 (Roche, Basel, Switzerland) [2]. The reaction mixture was composed of forward primer (10 μM), reverse primer (10 μM), 2 × TransStart^®^ Top Green qPCR SuperMix, and nuclease-free water. The measured Ct values were converted to relative copy numbers using the 2−ΔΔCt method. Three fully independent biological replicates were obtained and subjected to qPCR runs in technical triplicates.

WGCNA was performed as described previously based on the reads mapped through the RNAseq-workflow pipeline [2,48]. The gene modules were visualized using Cytoscape 3.9.1 [49]. ClueGo [50] and cytoHubba [40] built in Cytoscape were used to annotate the molecular functions and extract the hub genes from the genes in the modules. The relevant sequences were retrieved from https://www.kegg.jp (accessed on 1 October 2022). and https://phytozome-next.jgi.doe.gov (accessed on 1 March 2020).

### 4.5. Application of Exogenous GA_3_

To determine if the phenotype of *Gmssp* could be rescued by exogenous GA_3_ treatment, *Gmssp* and WT plants were grown in Hoagland nutrient solution using the hydroponic box method in a greenhouse [51]. Each treatment consisted of 10 plants. GA_3_ solution at a concentration of 0.1 μM was applied to the whole plant every other day for a total of four times, starting from the V1 stage (first trifoliate leaf fully open); water was used as the control [52].

### 4.6. Subcellular Localization

The *Glyma.11G026400* coding sequence was cloned from WT and *Gmssp* plants using primer pairs GFP-399 and GFP-399-4bp and fused in-frame into the expression vector p35S-GFP for subcellular localization. The constructs were transformed into *Agrobacterium tumefaciens* and co-infiltrated into the leaves of 4 week old *Nicotiana benthamiana* plants. The p35S-GFP empty vector was used as a control [22]. The *N. benthamiana* plants were grown under long-day (16-h light/8-h dark) conditions. Leaves were observed 3 days after infiltration using the Zeiss LSM700 scanning laser confocal microscope and image software (Zen 2011, Carl Zeiss MicroImaging GHBH, Jena, Germany).

## 5. Conclusions

Using the BVF-IGV pipeline established in this study, we identified *Glyma.11G026400* as the causal gene for the *Gmssp* phenotype, including super-short petioles and enlarged petiole angles. QTL mapping, analysis of a multi-gene deletion mutant, GA treatment, and WGCNA revealed the roles of *Glyma.11G026400* in cell division, cell differentiation, and cell enlargement in a tissue-dependent manner via the GA and APC pathways (Figure 8). Further functional characterization of GmAPC8 might enable us to use allelic variations or targeted mutations to develop soybean plants with ideal plant architecture to enhance soybean yield.

## Figures and Tables

**Figure 1 ijms-24-11024-f001:**
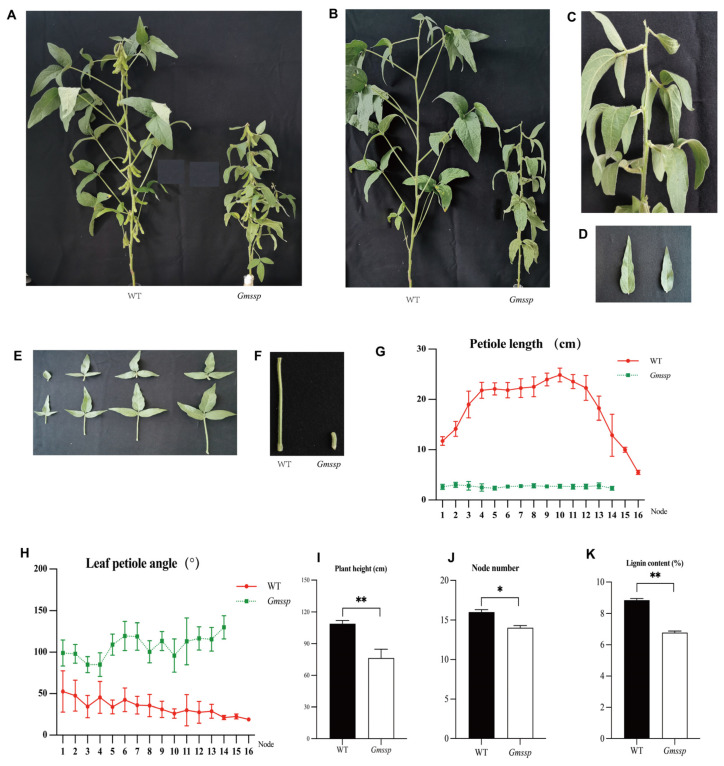
The *Gmssp* mutant displays super-short petioles and enlarged petiole angles at the vegetative stage. (**A**,**B**) Plant architecture (**A**) and plant architecture after all pods were removed (**B**) of *Gmssp* and the wildtype (WT). (**C**) Close-up view of the *Gmssp* mutant showing enlarged petiole angles. (**D**–**F**) Leaf (**D**), tri-leaflet (**E**), and petiole (**F**) size in *Gmssp* and WT. (**G**,**H**) Differences in petiole length (**G**) and petiole angle (**H**) at different nodes between *Gmssp* and WT. (**I**–**K**) Differences in plant height (**I**), node number (**J**), and lignin content (**K**) between *Gmssp* and WT. *, ** Significant differences at *p* < 0.05 and *p* < 0.01, respectively.

**Figure 2 ijms-24-11024-f002:**
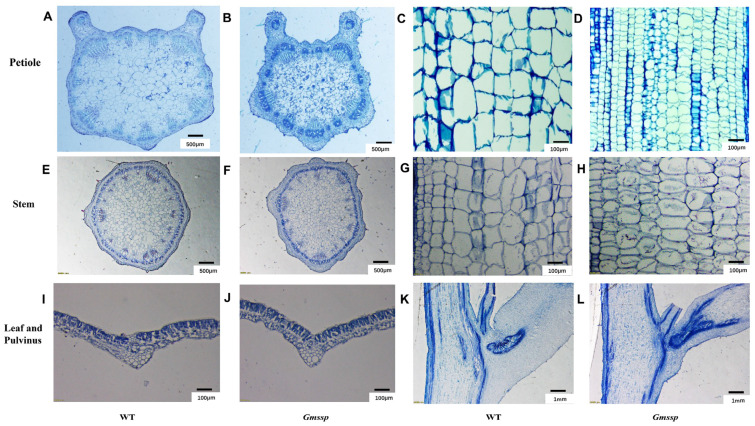
*Gmssp* shows anatomical differences from the WT in several tissues. (**A**,**B**) Transverse sections of a petiole in WT (**A**) and *Gmssp* (**B**). (**C**,**D**) Longitudinal sections of a petiole in WT (**C**) and *Gmssp* (**D**). (**E**,**F**) Transverse sections of a stem in WT (**E**) and *Gmssp* (**F**). (**G**,**H**) Longitudinal sections of a stem in WT (**G**) and *Gmssp* (**H**). (**I**,**J**) Transverse sections of a leaf in WT (**I**) and *Gmssp* (**J**). (**K**,**L**) Longitudinal sections of a pulvinus in WT (**K**) and *Gmssp* (**L**).

**Figure 3 ijms-24-11024-f003:**
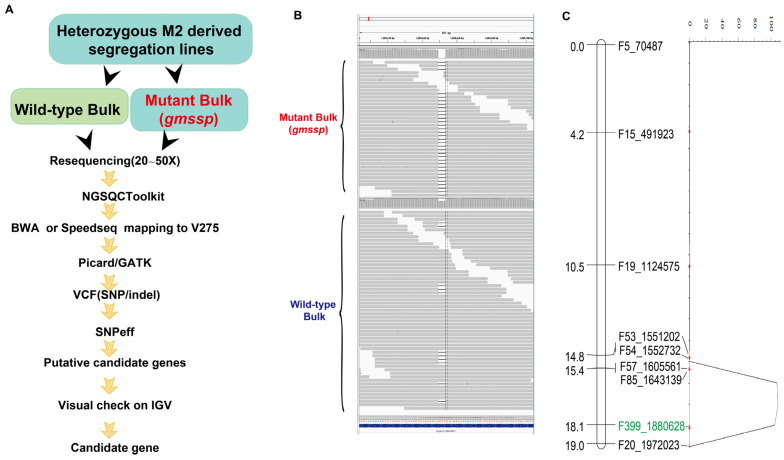
Bulk sequencing, variant calling, functional annotation using SnpEFF software, and the Integrative Genomics Viewer (BVF-IGV) pipeline for identifying the candidate gene for *Gmssp.* (**A**) Diagram of the BVF-IGV pipeline. (**B**) The allelic variations of the candidate gene *Glyma.11G026400* were manually checked on IGV. (**C**) The QTL peaks were detected in the candidate gene-anchored region using an F_2_ population of *Gmssp* × Hefeng 55.

**Figure 4 ijms-24-11024-f004:**
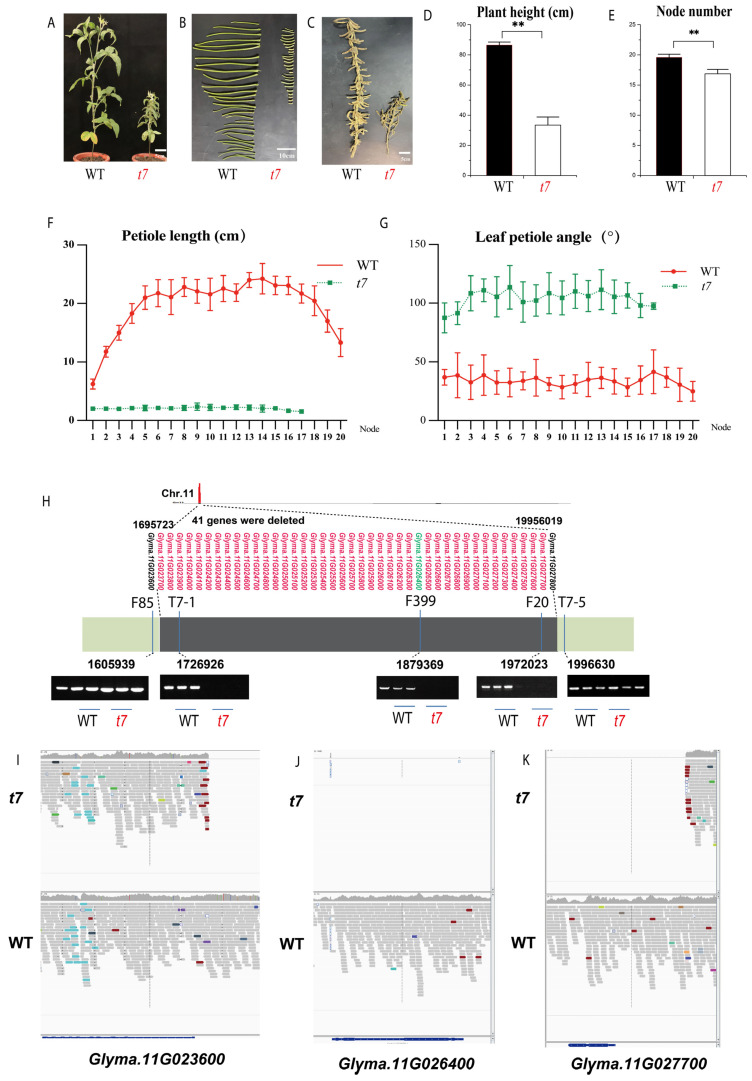
The *t7* mutant, with a large deletion involving many genes, including *Glyma.11G026400*, is phenotypically similar to the *Gmssp* mutant. (**A**–**C**) Plant architecture (**A**), petioles (**B**), and mature seed pods (**C**) of *t7* and Dongsheng 7 (WT). (**D**,**E**) Plant height (**D**) and node number (**E**) of *t7* and WT. ** Significant differences at *p* < 0.01. (**F**,**G**) Petiole length (**F**) and petiole angle (**G**) at different nodes of *t7* and WT. (**H**) Confirmation of a large deletion from Chr11:1697212 to Chr11:1985099 in which 41 genes were deleted, including the entire *Glyma.11G026400* gene. The physical locations of the markers are indicated. (**I**–**K**) Snapshots from IGV showing the left side (**I**), at *Glyma.11G026400* (**J**), and the right side (**K**) of the large deletion in *t7* compared to the WT. Alignments whose mate pairs were mapped to unexpected locations are color-coded according to the chromosome of the mate; other alignments are shown in light gray.

**Figure 5 ijms-24-11024-f005:**
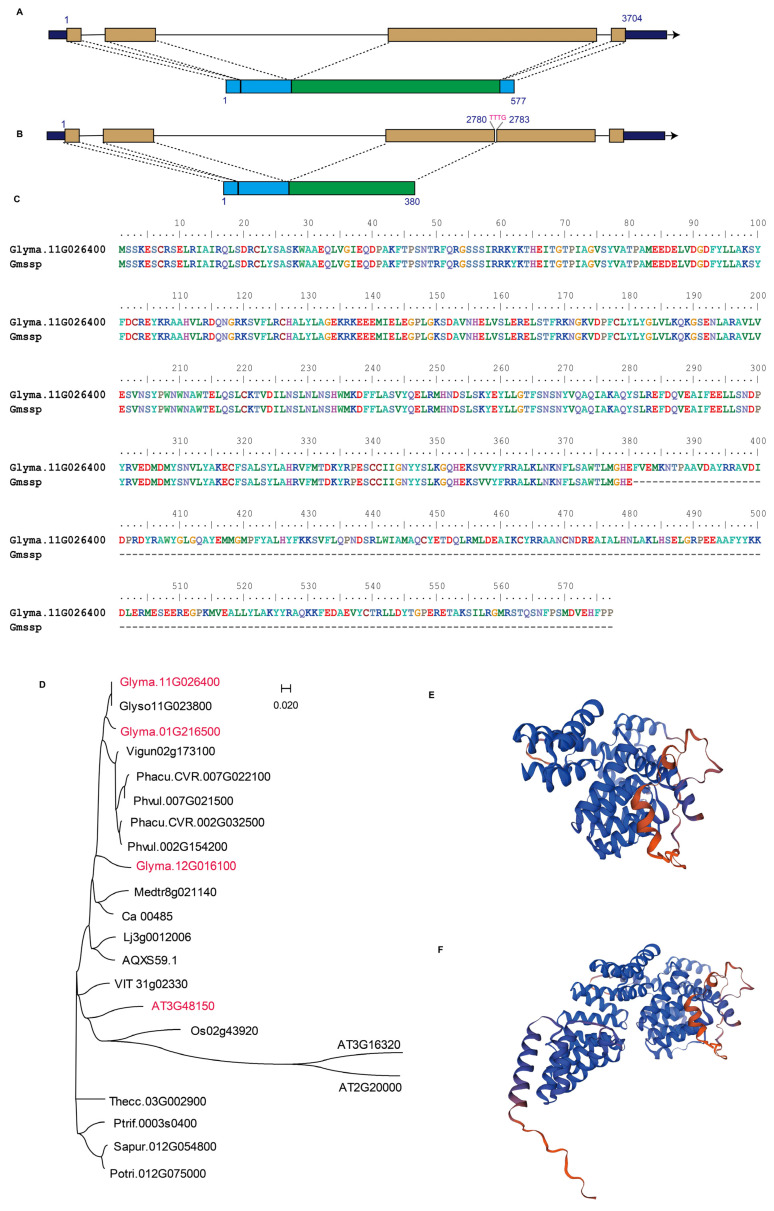
Characterization of the candidate gene, *Glyma.11G026400*. (**A**,**B**) The gene structure (**upper**) and putative protein structure (**lower**) in the WT (**A**) and the *Gmssp* mutant (**B**). Brown bars and black lines respectively represent coding sequence and introns, while black bars represent UTR region. The 4-bp deletion was marked in red. In the protein structure, the region corresponding the third exon where the deletion is occurred is marked in green. (**C**) Alignment of the APC8 protein sequence from the WT and *Gmssp*. The 4 bp deletion in the third exon of *Glyma.11G026400* leads to a truncated protein in *Gmssp*. (**D**) Phylogenetic tree of APC8 protein sequences from leguminous species and two model plants, *Arabidopsis* and rice. All sequences were retrieved from Phytozome (v13) and were aligned using the ClustalW program in BioEdit (v7.0.5.3) software (**E**,**F**) predicted 3D protein structure of APC8 in WT (**E**) and *Gmssp* (**F**) using the SWISS-MODEL online modeling server and visualized with PyMOL.

**Figure 6 ijms-24-11024-f006:**
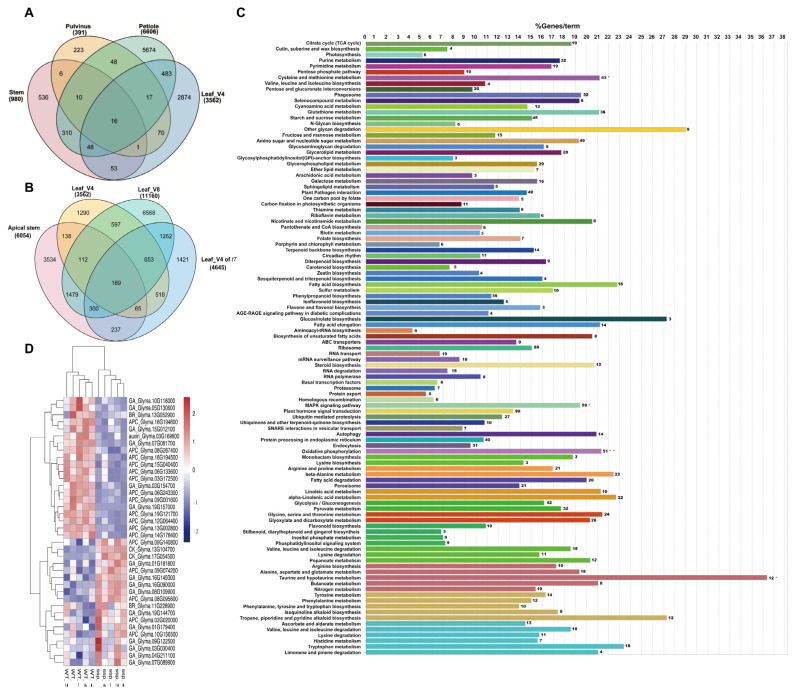
Transcriptome analysis of the *Gmssp* mutant and the WT. (**A**,**B**) Venn diagrams showing the number of differentially expressed genes (DEGs) between *Gmssp* or *t7* and their respective WTs that are shared among different tissues; the total number of DEGs is indicated under the tissue name. (**A**) Stem, pulvinus, petiole, and leaf at the V4 stage. (**B**) Apical meristem, leaf at the V4 stage, leaf at the V8 stage, and leaf of *t7* at the V4 stage. (**C**) Gene Ontology (GO) term and pathway enrichment analysis among the DEGs between the WT and *Gmssp* in petiole tissue at the V4 stage using ClueGO v2.5.9. The names on the left (y axis) represent the enriched GO terms or pathways; the numbers at the top (x axis) represent the percentage of genes per term; the number of enriched genes is indicated at the end of each bar. GO terms or pathways of the same color next to each other belong to the same group according to KappaScore grouping. *, ** Significant differences at *p* < 0.05 and *p* < 0.01, respectively. (**D**) Heat map of genes involved in APC-related ubiquitin-mediated proteolysis (APC) and plant hormone signal transduction pathways, i.e., gibberellin (GA), cytokinin (CK), auxin, and brassinosteroid (BR), in petioles. Sample names are indicated at the bottom; the related pathway abbreviation is listed before the gene name.

**Figure 7 ijms-24-11024-f007:**
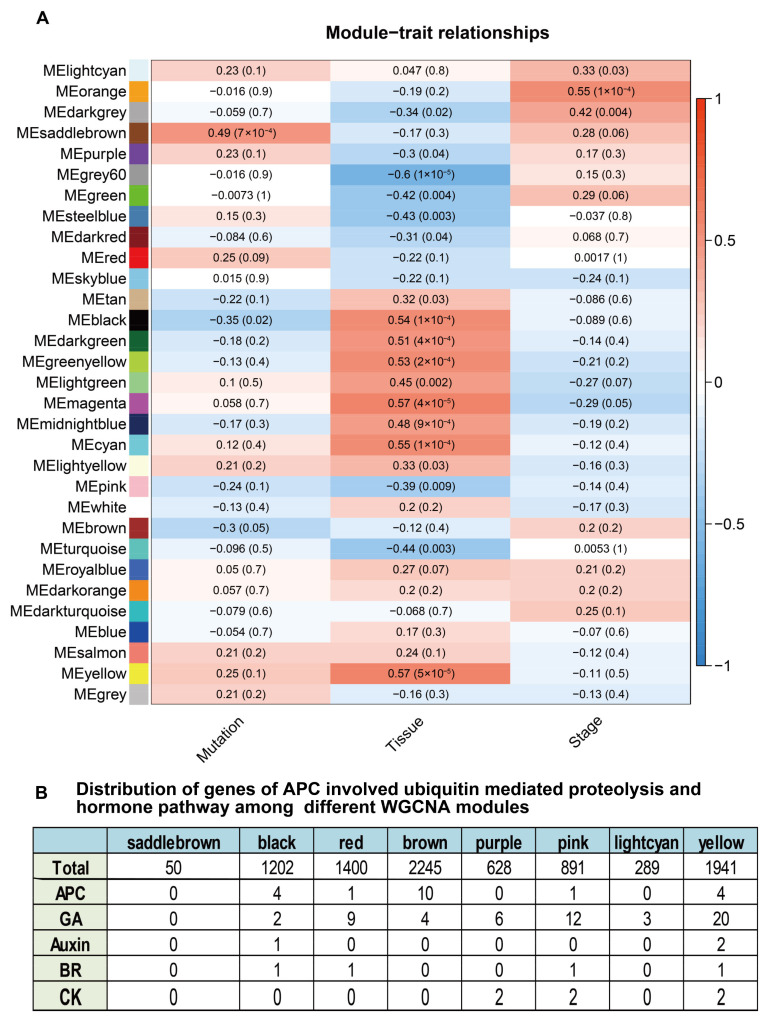
Relationships between modules and traits revealed by WGCNA. (**A**) The relationships between modules and traits of mutations, tissues, and developmental stages. Module names are listed on the left and traits at the bottom. In the table, data are presented as correlation coefficients with *p*-values in brackets. (**B**) distribution of genes involved in APC-related ubiquitin-mediated proteolysis and plant hormone signal transduction pathways among different WGCNA modules. APC, GA, auxin, BR, and CK represent the APC-related ubiquitin-mediated proteolysis, gibberellin, auxin, brassinosteroid, and cytokinin pathways, respectively.

**Figure 8 ijms-24-11024-f008:**
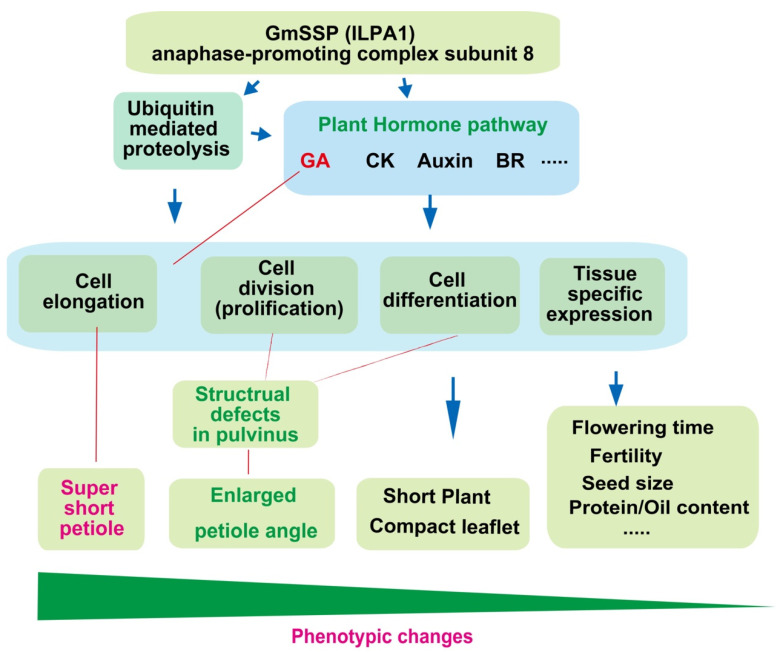
Diagram of the proposed functional mechanism of GmSSP. The dysfunction or absence of this protein affects many pathways, e.g., ubiquitin-mediated proteolysis and plant hormone signal transduction (GA, CK, auxin, and BR). Hence, developmental processes, such as cell division, cell differentiation, and cell elongation, are severely affected in a tissue-specific manner, affecting a variety of tissues and growth processes in different ways. Super-short petioles might result from dysfunction of the GA signaling pathway.

## Data Availability

The original data presented in the study are publicly available. The raw sequencing data reported in this article were deposited in the Genome Sequence Archive (Genomics, Proteomics & Bioinformatics 2021) at the National Genomics Data Center (Nucleic Acids Res 2021), China National Center for Bioinformation/Beijing Institute of Genomics Chinese Academy of Sciences (GSA: CRA009694 for RNA-seq of all tissues of *Gmssp* and WT L399, CRA010628 for both the resequencing data and RNA-seq data of *t7* and WT, CRA010631 for resequencing data of *Gmssp* bulk and WT bulk L399) and are publicly accessible at https://ngdc.cncb.ac.cn/gsa.

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
