# Peer review of "Dysfunction of an Anaphase-Promoting Complex Subunit 8 Homolog Leads to Super-Short Petioles and Enlarged Petiole Angles in Soybean"

_ijms, 2023, doi:10.3390/ijms241311024_

Round 1

Reviewer 1 Report (Previous Reviewer 2)

As I said in the evaluation of the previous form of the work, the size and position of the leaves is very important in all crop plants for increasing the leaf surface per unit of soil surface, for increasing the potential of the plants to produce more.

The results of this study contribute to the clarification of the mechanism that can influence the appearance (architecture) of plants, in the case of soybean, and can help researchers in creating new performing varieties of this very important species.

The purpose of the research should be clearly highlighted, before chapter 2. "Results"

Correction suggestions:

Line 57: instead of "Clark et al.",    better Clark et al. (2022).   Likewise in L. 59 instead of "Zhang et al.", Zhang et al. (2022).      Ditto for other similar cases.

Author Response

Dear Editor:

Thank you for giving us to revise our manuscripts. We have edited the entire manuscript according to all comments accordingly. Especially, the current version has been edited by the English editing service (Plant Editors http://planteditors.com).

We do hope the current version can be acceptable.

Here are the responses to each comments.

Reviewer 1

As I said in the evaluation of the previous form of the work, the size and position of the leaves is very important in all crop plants for increasing the leaf surface per unit of soil surface, for increasing the potential of the plants to produce more.

The results of this study contribute to the clarification of the mechanism that can influence the appearance (architecture) of plants, in the case of soybean, and can help researchers in creating new performing varieties of this very important species.

The purpose of the research should be clearly highlighted, before chapter 2. "Results"

Author’s response: Thanks for your valuable comments. We revised accordingly, and added some new sentences, i.e. “The knowledge yielded in this study would help breeders in creating new performing soybean cultivars having new appearance (architecture) and higher yield potentials.”

Correction suggestions:

Point: Line 57: instead of "Clark et al.", better Clark et al. (2022).   Likewise in L. 59 instead of "Zhang et al.", Zhang et al. (2022).  Ditto for other similar cases.

Author’s response: Thank for your carefulness. We revised per your suggestion throughout the manuscript.

Reviewer 2 Report (New Reviewer)

Kindly find the comments at the attachment

Author Response

Dear Editor:

Thank you for giving us to revise our manuscripts. We have edited the entire manuscript according to all comments accordingly. Especially, the current version has been edited by the English editing service (Plant Editors http://planteditors.com).

We do hope the current version can be acceptable.

Here are the responses to each comments.

.

Reviewer 2

Dysfunction of an anaphase-promoting complex subunit 8 homolog leads to super short petioles and enlarged petiole angles in soybean.
Yi Gao, Jinlong Zhu, Hong Zhai, Kun Xu, Xiaobin Zhu, Hongyan Wu, Wenjing Zhang,
Shihao Wu, Xin Chen, Zhengjun Xia

Point 1: Lines 488-489: Leaf, petiole, pulvinus, stem, and apical meristem tissues in Gmssp mutants and WT plants were sampled for RNA sequencing.

Please explain, what does mean “mutants and WT plants were sampled for RNA sequencing”. One sample for leaf, one sample for petiole, etc...? Probably Table S9.

Author’s response: Leaf, petiole, pulvinus, stem, and apical meristem tissues from Gmssp and WT plants were sampled for RNA sequencing. The basic information on genotype, tissue, sampling time (at different growth stage) are listed in Table S9.

Point 2: Lines 524-525: To verify the BVF-IGV results, Gmssp was crossed with Hefeng 55 to generate a QTL mapping population.
Please explained, how the population was created (SSD?) and samples from which generation were analyzed? How the DNA was isolated?

Author’s response: To verify the BVF-IGV results, the homogenous Gmssp (male) at M4 generation was crossed with Hefeng 55 (female) to generate a QTL mapping population. In the F2 population, DNA was extracted using CTAB method [46] Markers for QTL mapping in the putative candidate gene-anchored region were developed by targeting the polymorphisms between Gmssp/Heihe 43 and Hefeng 55 based on the re-sequencing data of Gmssp, Heihe 43, and Hefeng 55 (Table S5). Generally, the InDel variants were used to develop InDel markers, while SNP variants were used to develop CAPS/dCAPS. The software QTLIciMapping version 4.0 [47] was used for QTL analysis.

Point 3: Lines 525-527: Markers for QTL mapping in the putative candidate gene-anchored region were developed by targeting the polymorphisms between Gmssp/Heihe 43 and Hefeng 55 based on the re-sequencing data of Gmssp, Heihe 43, and Hefeng 55.
Please write more clear, what does mean “markers were developed by targeting the polymorphisms based on the re-sequencing data”.

Author’s response: Markers for QTL mapping in the putative candidate gene-anchored region were developed by targeting the polymorphisms between Gmssp/Heihe 43 and Hefeng 55 based on the re-sequencing data of Gmssp, Heihe 43, and Hefeng 55 (Table S5). Generally, the InDel variants were used to develop InDel markers, while SNP variants were used to develop CAPS/dCAPS.

Point 4: Lines 541-542: TransStart Top Green qPCR SuperMix (AQ131-04; TransGen Biotech, Beijing, China) was used for the qPCR assays. qPCR was conducted using a LightCycler 96 (Roche) [2].
Please add reagents and reaction mixtures for qPCR.

Author’s response: We have revised accordingly per your suggestion.Reverse transcription quantitative PCR (RT-qPCR) was performed to validate the RNA-seq result. The total RNA of different tissues was extracted using an OminiPlant RNA Kit (DNase I) (CW25985, CEBIO). A total of 500 ng RNA was used for reverse transcription by TransScript One-Step gDNA Removal and cDNA Synthesis SuperMix (AT311-03, TransGen Biotech). TransStart Top Green qPCR SuperMix (AQ131-04; TransGen Biotech, Beijing, China) was used for the qPCR assays. qPCR was conducted using a LightCycler 96 (Roche) [2]. The reaction mixture was composed of forward primer (10μM), reverse primer (10μM),2×TransStart® Top Green qPCR SuperMix, nuclease-free water. The measured Ct values were converted to relative copy numbers using  the 2−ΔΔCt method[17].

Point 5: Lines 542-543: The measured Ct values were converted to rela?ve copy numbers using the −ΔΔCt method.
Please check the correctness and add the reference.

Author’s response: The measured Ct values were converted to relative copy numbers using the 2−ΔΔCt method [17].

Point 6: Line 99: Figure 1. The Gmssp mutant displays super short petioles and enlarged petiole angles at the vegetative stage. differences in petiole angle (H).
Please add in the Material and Methods section how the angles were measured? Also how the petiole length was measured?

Please add *, **, *** description.

Author’s response: The average petiole length in the M4 population was 2.68 ± 0.52 cm in the Gmssp mutants and 19.51 ± 5.20 cm in the WT; the average petiole length of Gmssp was 13.74 % that of the WT (P < 0.001; Figure 1G). Gmssp also had significantly larger angles between the main stem and petiole (106.51 ± 19.78 °) than the WT (34.86 ± 3.05 °) (P < 0.001; Figure 1H).We added “*, ** represent statically significant difference at P <0.05 and P < 0.01, respectively.

Point 7: Line 115: had significantly smaller leaves 115 (19.11 ± 4.0 cm2), Please describe, how the leaf area was measured?

Author’s response: For phenotypying, the length of leaf petiole in the main stem was measured by a ruler, while the angle between the main stem and the petiole was measured by a protractor. For measuring the leaf area, the leaf was photographed along with the horizontal and vertical rules, the leaf area was measured with aid of the Image J software. At R7 stage, plant heights were measured from the cotyledon node to the shoot tip of the main stem using a ruler. Seed protein and oil contents of each plant or genotype were measured using MATRIX-I FTNIR spectrometer (Bruker).

Point 8: Line 117-122: Furthermore, Gmssp had a lower oil content (18.10 ± 0.72 %) and higher protein content (42.63 ± 0.15 %) than the WT (20.50 ± 0.44 % for oil content and 41.40 ± 0.40 % for protein content) (P < 0.01; Figure S1). Also, Gmssp had a lower lignin content (6.78 ± 0.09 %) than the WT (8.85 ± 0.11 %) at P < 0.01 (Figure 1K). In addition, Gmssp had lower neutral detergent fiber (NDF), acid detergent fiber (ADF), and crude fiber (CF) contents than the WT (Figure S1).
Please add in the Material and Methods section how the oil, protein contents were assessed.

Author’s response: Seed protein and oil contents of each plant or genotype were measured using MATRIX-I FTNIR spectrometer (Bruker).

Point 9: Lines 166-167: After cleaning, we aligned the bulk sequences of four Gmssp mutants and six WT plants to the reference genome Gmax_275_Wm82.a2.v1 (V275) using BWA software [20].

What does it mean “after cleaning”?

Author’s response: We changed accordingly. “Reads were quality-trimmed by the NGS QC Toolkit (v2.3.3) (Patel and Jain, 2012) with default parameters”

Point 10: Lines 177-179: Since some genes contained more than one functional allelic variation, we subjected 1,760 genes specific to Gmssp and 2,030 specifics to the WT to batch IGV observation using a script. Please transform this sentence to be more understandable.

Author’s response: By manipulation of VCF data in Excel, we eliminated the common allelic variations between the Gmssp and WT bulks, leaving 3,964 loci (1,760 genes) specific to Gmssp and 4,922 loci (2,030 genes) specific to the WT. Then these genes were manually checked individually or in batch by taking snapshots of each gene into a fold using the function of “Run Batch Script” built in IGV.

Point 11: Lines 182-184: Initially, we used a script to take a snapshot for each of the 1,760 Gmssp specific genes on IGV to scan for allelic variations.
Could you describe this more biological?

Author’s response: By manipulation of VCF data in Excel, we eliminated the common allelic variations between the Gmssp and WT bulks, leaving 3,964 loci (1,760 genes) specific to Gmssp and 4,922 loci (2,030 genes) specific to the WT. Then these genes were manually checked individually or in batch by taking snapshots of each gene into a fold using the function of “Run Batch Script” built in IGV.

Point 12: Lines 184-187: since the allelic variations were in good accordance with the prediction that about 1/6 of the reads were Gmssp type, given that half of the plants with a WT phenotype from the heterozygote derived segregation line were assumed to be heterozygous at Gmssp (Figure 3B). Please explain more clearly.

Author’s response: After an initial check, we identified a 4-bp deletion in Glyma.11G026400, marking this gene as a strong candidate for Gmssp. Interestingly, the Gmssp bulk showed homogenous deletion for all reads, while approximately 1/6 reads of WT bulk contained the 4-bp deletion, which were in good accordance with the fact that half of WT plants were heterozygous at the Gmssp locus.

Point 13: Lines 201-203: We identified polymorphisms based on the re-sequencing data of the parents, Gmssp (in the Heihe 43 background) and Hefeng 55, and developed corresponding markers in the region where the candidate gene is anchored.
Could you describe more clear what have you done?

Author’s response: We identified polymorphisms (inDels and SNPs) based on the re-sequencing data of the parents, Gmssp (in the Heihe 43 background) and Hefeng 55, and developed corresponding inDels and SNPs markers in the region where the candidate gene is anchored. Then, we performed QTL analysis for Gmssp (super short petiole and enlarged petiole angle) (Table S2), which showed that the QTL peak overlapped with the candidate gene-anchored region (Figure 3C). We analyzed the allelic variations among Gmssp, WT (Heihe 43 background), and Hefeng 55 within the QTL peak region, finding that a 4-bp deletion leading to a premature stop codon in Glyma.11G026400 in Gmssp was the only mutation in this region (Table S3). These QTL mapping results strongly supported our BVF-IGV pipeline results (Table S3). We described in detail in the M and M section

Point 14: Lines 203-204: Then, we performed QTL analysis (Table S2), which showed that the QTL peak overlapped with the candidate gene-anchored region (Figure 3C).
Please explain what does 0, 1, 2, -1 mean in the Table? Which trait was mapped? Gmssp phenotype generally or super short petiole and enlarged petiole angles separately?

Author’s response:According to the manual of the software of QTLici mapping, AA genotype ---2, AB ---1, BB –0  Missiing, -1,  the Scheme 1, one of three commonly adopted schemes.  We revised the Talbe S2 accordingly. Here, we only did QTL for Gmssp, Gmssp (super short petiole and enlarged petiole angle), but not petiole length and petiole angle separately, since we did not found any contradiction between petiole length and petiole angle.

Point 15: Lines 221-222: Figure 4. The t7 mutant, with a large deletion involving many genes, including Glyma.11G026400, is phenotypically similar to the Gmssp mutant. Plant height (D)
Please add in the Material and Methods section how the plant length was measured?

Author’s response: At the R7 stage, plant heights were measured from the cotyledon node to the shoot tip of the main stem using a ruler.

Point 16: Lines226-227: I–K, Snapshots from IGV showing the left side (I), at Glyma.11G026400 (J), and the right side (K) of the large deletion in t7 compared to the WT.
Please explain what do the colors on the picture (I), (J), (K) mean?

Author’s response: Alignments whose mate pairs are mapped to unexpected locations are color-coded by the chromosome of the mate; other alignments are displayed in light gray.We indicated in the legend of the Figure. Since these color-coded reads are relatively rare, which might have no impact on our conclusion.

Point 17: Lines 232-234: We employed the primers used for QTL mapping and designed new primers to validate the existence of this deletion (Table S5).
Could you describe more clear what have you done?

Author’s response: We employed the primers of F85, F399, and F20 used for QTL mapping and newly designed primers (T7-1 and T7-5) using Primer3 software to validate the existence of this deletion (Chr11:1695723 to Chr11:1995601) (Table S5).

Point 18: Table S4: Genes expressed in the Gmssp target region analyzed using the RNAseq-workflow pipeline. Please add the explanation of values in Table S4 (units, relative quantification from qPCR, data from RNAseq?). If relative quantification data, please explain the formula calculation Ct value to relative quantification.

Author’s response: Thank you for your kind suggestion. In Table S4, these value are reads yielded after

We performed RNA sequencing (RNA-seq) and 150-bp pair end resequencing of genomic DNA from a mutant bulk of 20 t7 plants and a WT bulk of 20 plants (Table S4). For RNA-seq, we detected no expression for 41 genes within the Glyma.11G026400-anchored region in the t7 bulk, suggesting that a large deletion occurred in this region (Table S4).

(Table S4. Genes expression in the Gmssp target region using the RNAseq-workflow pipeline.

Note: Data in this table represent the reads yielded through analysis of RNA seq data using the RNAseq-workflow pipeline.)

Point 19: Line 234: The PCR results confirmed the lack of gene expression in t7 in this region Please add in the Material and Methods section the PCR reagents, mixtures and profile.

Author’s response: Thank you for your kind suggestion. We revised per your suggestion throughout the manuscript.

Point 20: Lines 279-280: We manually checked the authenticity of the genotype at the GmAPC8 locus for each sample on IGV.

Could you describe more clear what have you done?

Author’s response: The reaction mixture was composed of forward primer (10μM), reverse primer (10μM),2×TransStart® Top Green qPCR SuperMix, nuclease-free water. The measured Ct values were converted to relative copy numbers using the 2−ΔΔCt method [17].

Point 21: Line 356: Figure 7. B, distribution of genes involved in APC-related ubiquitin-mediated proteolysis In the table, tatal?

Author’s response: Thank you very much for your careful correction. We changed accordingly.

Point 22: Lines 445-446: but did not affect reproductive development, e.g., flowering time and seed protein + oil content.
But the only oil content, yes, so why the authors choose the seed protein + oil content parameter?

Author’s response: Since the protein and oil contents were individually different between Gmssp and WT, but important trait of protein + oil content was not significantly different.

Point 23: Lines 469-471: The outer cells of the parenchyma, termed the “motor cells”, are responsible for nyctinastic and thigmotactic movement through water-driven volume changes (Moran, 2007). Please change the citation style. Lack of this position in the reference.

Author’s response: Thank you very much for your careful correction. We changed accordingly.

General comments:
I. The Results part is very extensive, has many aspects.

  1. The Gmssp mutant has a dwarf phenotype with a super short petiole and enlarged petiole angles. petiole length, angle, leaves area, oil, protein, lignin, fiber content, 100-seed weight
  2. anatomical differences between Gmssp and WT tissues.
  3. The BVF-IGV pipeline identified Glyma.11G026400 as the candidate gene for the Gmssp phenotype
  4. QTL mapping validated Glyma.11G026400 as the candidate gene for the Gmssp phenotype Cross the Gmssp mutant with Hefeng 55,
  5. Characterization of the t7 mutant supported Glyma.11G026400 as the causal gene for the Gmssp phenotype Plant height, nodes number, petiole length, angle, qPCR of 41 genes,
  6. 11G026400 encodes a truncated protein in the Gmssp mutant a phylogenetic tree
  7. The mutation in GmAPC8 in the Gmssp mutant does not alter its subcellular localization
  8. Enrichment of ubiquitin-mediated proteolysis and hormone signal transduction

RNA sequencing of petiole, pulvinus, leaf, stem, and apical meristem tissues and analyzed the data using the RNA-seq workflow pipeline
differentially expressed genes (DEGs)

  1. Gibberellin treatment partially rescued the phenotype of the Gmssp mutant
  2. WGCNA revealed modules associated with both the tissue and growth stage traits of Gmssp

Comparing with the Results part, the Discussion is quite short. Especially I would like to see the connection between very broad results part regarding DEGs in the different variants and the aim of the research.

  1. Short petioles and enlarged leaf petiole angles are important traits determining plant architecture in soybean.
  2. The BVF-IGV pipeline facilitated the identification of the causal mutation for the Gmssp phenotype
  3. 11G026400 may influence soybean plant architecture via ubiquitin-mediated proteolysis (APC) and plant hormone signal transduction
  4. I have the impression that the authors use many mental shortcuts ei.

Point: Lines 201-203: We identified polymorphisms based on the re-sequencing data of the parents, Gmssp (in the Heihe 43 background) and Hefeng 55, and developed corresponding markers in the region where the candidate gene is anchored.
Could you describe more clear what have you done?

Author’s response: We identified polymorphisms (inDels and SNPs) based on the re-sequencing data of the parents, Gmssp (in the Heihe 43 background) and Hefeng 55, and developed corresponding inDels and SNPs markers in the region where the candidate gene is anchored. Then, we performed QTL analysis for Gmssp (super short petiole and enlarged petiole angle) (Table S2), which showed that the QTL peak overlapped with the candidate gene-anchored region (Figure 3C). We analyzed the allelic variations among Gmssp, WT (Heihe 43 background), and Hefeng 55 within the QTL peak region, finding that a 4-bp deletion leading to a premature stop codon in Glyma.11G026400 in Gmssp was the only mutation in this region (Table S3). These QTL mapping results strongly supported our BVF-IGV pipeline results (Table S3). We described in detail in the M and M section. Furthermore, the DEGs varied widely among different tissues, which might help us to understand why Glyma.11G026400, the APC8 homolog playing key roles in cell differentiation and elongation in a tissue-dependent manner.

III. Some informations are hard to understand due to use the terms like “ use a script to take a snapshot” or “ batch IGV observation using a script”

Point: Lines 177-179: Since some genes contained more than one functional allelic variation, we subjected 1,760 genes specific to Gmssp and 2,030 specifics to the WT to batch IGV observation using a script.

Author’s response: By manipulation of VCF data in Excel, we eliminated the common allelic variations between the Gmssp and WT bulks, leaving 3,964 loci (1,760 genes) specific to Gmssp and 4,922 loci (2,030 genes) specific to the WT. Then these genes were manually checked individually or in batch by taking snapshots of each gene into a fold using the function of “Run Batch Script” built in IGV.

Point: Lines 182-184: Initially, we used a script to take a snapshot for each of the 1,760 Gmssp specific genes on IGV to scan for allelic variations.
Could you describe this more biological?

Author’s response: By manipulation of VCF data in Excel, we eliminated the common allelic variations between the Gmssp and WT bulks, leaving 3,964 loci (1,760 genes) specific to Gmssp and 4,922 loci (2,030 genes) specific to the WT. Then these genes were manually checked individually or in batch by taking snapshots of each gene into a fold using the function of “Run Batch Script” built in IGV.

This manuscript is a resubmission of an earlier submission. The following is a list of the peer review reports and author responses from that submission.

Round 1

Reviewer 1 Report

In this paper, the improved BSA (Bulked-segregate analysis) was used to locate the gene Glyma.11G026400, which controls the important characters (leaf petiole and petiole angles) of soybean, and other methods and materials were used to verify the Glyma.11G026400 gene, and the molecular basis of Glyma.11G026400 gene mutation was analyzed in depth, which is a very valuable work.

There are some small problems and suggestions. The author can consider making some modifications.

A large number of differential genes were found in the transcriptome analysis of 2.7. Transcriptome Analysis, but the author did not analyze the relationship with these differential genes around Glyma.11G026400. This gene was not even found in Figure S7 and Table S7.

Here, the author believes that the kegg pathways such as APC, Auxin, CK, BR, and GA are very important, but the results show that these kegg pathways are not enriched (there is no significance test result in the article, only the number of genes is listed in the table). Can we consider other kegg pathways, such as  the common DEG or metabolic pathway of these tissues. The results obtained here need not be consistent with other studies, which can be further discussed. 

The author believes that soybean petiole length and leaf angle are recessive genes controlled by a single gene, and even the result of 3:1 was obtained in the QTL analysis (there is no intermediate form here). But in the discussion, we found that there are other genes controlling this trait, and this trait is not like a quality trait. How much role does Glyma.11G026400 gene play in controlling soybean petiole length and leaf angle? Can this question be clarified in the article, Or explain it in the discussion.

123-126 The three replicates are combined to conduct the overall statistical test to obtain a more definite result. If you get an unexpected result, you can explain it.

127-142 Some results give P values, and some do not.

139 The value of the lignin content of gmssp is missing 

232 16.87 (SD ± 2.28) ---- SD (16.87 ± 2.28), some results give P values, and some do not.

301 How are the genes in Table S7 selected? Does Table 1 represent the specific expression of a gene in this tissue?

301-304  There are no unanimous DEGs among all tissues(301) ------ Glyma.11G026400 was higher in WT than in gmssp mutants(304).   Is Glyma.11G026400 also not a DEG?

 304 (Figure S4). It should be Figure S5

Reviewer 2 Report

1. The objective of this work was to identify the gene that causes the mutation gmssp mutant with super short petiole and enlarged petiole angle in soybean.

2. The subject is important, because it relates to soybean genetics and the possibilities of improving its architecture and, implicitly, photosynthetic efficiency. Any addition to this knowledge is welcome. The authors describe in detail the correlation between the mutant gene components and the anatomy of soybean leaf petiole parts. The defects in anatomical structures are well associated with the dysfunction of the pulvinus of gmssp.

3. The presented results describe in detail the manifestation of the Glyma.11G027700 gene and reveal the key function of Glyma.11G026400 in cell division, differentiation, enlargement and tissue development through GA and APC pathways. The paper exhaustively presents the correlations between the mentioned gene and its effects on Short Petiole and Leaf Angle in soybean.

4. I have no suggestions for improving the paper and I believe that the methodology was correct.

5. References are appropriate to the topic studied. 7. I have no other comments on the paper.

Typing error

Line 361: after subchapter 3.1 follows subchapter 4.2. and other. Need to be corrected here.

Reviewer 3 Report

Line 463: please write in full first about M3 (mutant population generation 3)

line 485: how many individuals in each bulk, please mention

line 509: QTL again write in full first

line 486: these were the numbers of line? 399. please re-write and clear it

line 470-471: its confusing. you harveested a single line (399) or harvested the total 399 lines.

And please call them mutant inbred lines because you are selfing and you started selecting the single seed-based progeny from M2? Am I right?

line 485: refer bulks to the figure that shows the nice distribution of genotypes for targeted traits and then say you selected the tops-and-tails and made their bulks. It should be mentioned.

Also, please replace bulk sequencing with bulk segregant analysis

Snpeff reference

IGV reference

section 4.3: now the authors are mentioning the mapping population, read my above comment

section 5 should be after the discuyssion

conclusion needs to be revised. the authors repeated the results. Give something conclusive with future perpectives
